# Understanding Subordinate Animal Welfare Legislation in Australia: Assembling the Regulations and Codes of Practice

**DOI:** 10.3390/ani12182437

**Published:** 2022-09-15

**Authors:** Rochelle Morton, Alexandra L. Whittaker

**Affiliations:** School of Animal and Veterinary Sciences, The University of Adelaide, Roseworthy, SA 5371, Australia

**Keywords:** animal welfare legislation, animal cruelty, regulations, codes of practice, delegated legislation, animal welfare, Australia

## Abstract

**Simple Summary:**

In Australia, animal welfare protection is a state and territory responsibility. Having eight separate state and territories results in eight separate animal welfare legal frameworks, all which contain a primary piece of legislation and secondary or subordinate forms of legislation. These subordinate forms are known as regulations and codes of practice and are lower-ranking laws that are given powers under the primary legislation. Subordinate laws contain crucial details that govern everyday human–animal interactions and industry practices, for example companion animals used for breeding. There has been no review of the animal welfare-focused subordinate laws in Australia. This study has assembled each animal welfare regulation and code of practice in force in Australia as a reference for practitioners working in specific animal-related areas to easily identify relevant documents pertaining to welfare and understand the nature of their responsibilities in terms of compliance with these documents. A total of 201 pieces of subordinate legislation were identified and presented through the following utility categories of animals: companion, production, wild/exotic, entertainment. The benefits of housing the information identified from this study on an online database for animal industries to use are discussed.

**Abstract:**

The state-based approach to regulating animal welfare in Australia is thought to create national dis-uniformity in that each state and territory legislates and operates inconsistently. The animal welfare legal framework in each of the eight Australian jurisdictions is made up of a primary statute and subordinate legislation, where subordinate animal welfare legislation, in the forms of regulations and codes of practices, are lower-ranking laws that are given power under the jurisdiction’s specific animal welfare statute. Since a review of animal welfare statutes identified broad patterns between the jurisdictions, this study is intended to be complementary by collating the subordinate legislation to provide a more comprehensive understanding of animal welfare laws in Australia. Using targeted search strategies stemming from the eight enabling animal welfare statutes, this study identified 201 pieces of subordinate legislation in force between 28 March 2022 and 5 April 2022. The scope of subordinate legislation is depicted through the following utility categories of animals: companion, production, wild/exotic, entertainment. Whilst subordinate legislation differed between the jurisdictions, it was common for similar welfare concerns or topic areas to be protected in higher-order legislation (statutes or regulations). Additionally, many jurisdictions were found to have similar shortcomings, all which likely could be managed through a mechanism of national data collection.

## 1. Introduction

National animal welfare legislation is restricted in scope in Australia, as the *Commonwealth of Australia Constitution Act 1901* (‘Constitution’) [1] does not include a ‘head of power’ for animal welfare. Consequently, animal welfare protection is a residual power within the domains of the Australian states and territories, which have taken different approaches to legislating the issue [2,3,4,5,6]. As a result, the primary legal source of animal protection in Australia stems from eight individual animal welfare statutes made at the state level. This state-based approach has been argued to create a ‘fragmented, complex, contradictory, inconsistent system of regulatory management’ [3], on the basis that it causes public confusion [7], makes national data collection almost impossible [3], and does not present a united front toward animal protection [6]. For this reason, scholarly support for a more uniform approach to animal welfare legislation is commonplace [3,6,7,8] as in theory it should overcome such issues. There is, however, an argument proposed by Morton et al. [5] that national uniformity of written law is not necessary since animal welfare statutes have been shown to be broadly consistent between the jurisdictions, and to share similar shortcomings. Further, these identified shortcomings could likely be addressed without resort to a single overarching piece of national legislation [5]. However, that review was limited only to statutes, which do not work independently [9]; rather, they rely on a symbiotic relationship between subordinate legislation in the forms of regulations and codes of practices, in addition to common law principles. Morton et al. [5] stated that subsequent analyses of subordinate animal welfare laws are required to provide a comprehensive understanding on the extent of the cross-jurisdictional differences to animal welfare legislation.

Subordinate laws are lower-ranking laws, meaning they are enabled by a statute. They are often referred to as ‘delegated legislation’, as Parliament ‘delegates’ the authority of creating such legislation (i.e., regulations) to the executive arm of government [10]. Although they sit under statute, they still have the full force of the law just as a statute would [11]. Recent years have seen a significant shift from parliamentary law-making (i.e., statutes) to executive law-making (i.e., regulations), with 88% of new laws in 2020 being delegated laws just in the state of South Australia [12]. This gives some indication of the substantial mass of subordinate legislation in modern government today. The process of creating a statute requires substantial parliamentary oversight and debate, making it time-consuming and lengthy [10]. In contrast, as parliament delegates the authority to create regulations to the executive government, it eliminates the need for parliamentary oversight, making the process of creating and amending subordinate laws much quicker [13]. However, in order to avoid sole reliance on non-parliamentary law-making, a mechanism known as ‘disallowance’ has been developed to keep executive lawmakers in check, where parliament is notified of the delegated legislation and given the opportunity to disallow it [13]. Regulations are often used for more technical matters or matters of detail that are subjected to rapid changes, as it is more appropriate to house them in a form of legislation that can be changed with relative ease [14]. Additionally, regulations often house large amounts of detail to ensure that enabling statutes remain concise with clear provisions. In the case of animal welfare laws, regulations often detail provisions specific to the use of restricted equipment (electrical devices, traps), restricted surgical procedures (debarking, declawing) and detail minimum standards for specific industries (poultry, pigs) as just some examples.

Codes of practices (‘codes’) are another form of subordinate legislation. They are known as ‘quasi-delegated legislation’ or ‘soft law’ [10], a partial form of delegated legislation as they are often not drawn to the attention of parliament or subject to any form of disallowance [13]. Codes are guidance documents written for specific animal industries or utilities (commonly farming industries) that details the forms of ‘acceptable’ uses of animals [2] rather than solely relying on the ‘unacceptable’ forms detailed in statute [5]. They provide guidance to animal industries and allow ‘cruelty’ to be defined by specialists within industry in line with advancement in animal welfare science rather than relying on the sometimes-inconsistent interpretations of the judiciary [2,15].

Whilst regulations sit directly under their enabling statute (i.e., the statute that has authorized/enabled the delegation of legislative law-making power to the executive arm of government), codes are found lower in the hierarchy (Figure 1). This means that provisions kept in codes are often thought to be of lower weight than those in regulations, likewise for provisions found in regulations compared with statutes. Codes can either be compulsory or voluntary; those that are compulsory are legally enforceable, of a higher weight, and adopted under regulations through a direct reference, licensing requirements or inclusion in schedules. On the other hand, voluntary codes are lowest in the hierarchy (hence the lowest in weight) as they are not incorporated under regulations, giving industries discretion over whether they adopt them or not. As briefly mentioned, the enforceability of these forms of subordinate legislation can differ. Regulations are legally enforceable documents that have multiple offences detailed within the regulations for when breaches of various parts occur. In comparison, noncompliance with a code of practice is only punishable if the prescribing legislation (i.e., legislation higher in the hierarchy) includes an offence for noncompliance. In jurisdictions with any such offence, it only applies for a breach of provisions of compulsory codes, not voluntary codes. Voluntary codes are often only admissible in court, meaning they can only be used as a form of evidence in support or in defence of an animal welfare offence.

Subordinate legislation plays an important part in modern government in all common law countries, yet in a similar vein to animal welfare statutes, subordinate laws often come under scrutiny for their contribution to the inconsistent and contradictory framework of animal welfare legislation [2,6,13,16,17]. As summarized by Boom and Ellis [16], ‘the wide range of other legislative provisions and codes means the law lacks coherence and certainty’. This is because the Australian state and territory governments have discretion over their subordinate laws, resulting in large variation across the jurisdictions as to the animal species and practices addressed through regulations and codes [2]. Originally this was not the intention, as the Model Codes of Practice [18] for animal welfare were developed in the early 1980s to introduce some form of national consistency to farm animal practices [2]. However, each jurisdiction adopted the Model Codes differently; some adopted the codes in their entirety, others modified them, whilst some chose to ignore them completely [17]. Only after a federal report in 2005 shed light on the extent of the jurisdictional inconsistencies in animal welfare codes [19] did the federal government action a second attempt at national uniformity (amongst other issues identified in the report) with the development of the Australian Animal Welfare Standards and Guidelines [20]. As stated in 2013 by Dale and White [2], ‘it is likely to be many years before all existing Model Codes of Practice have been converted [to the standards and guidelines]’, and now almost a decade later, it would appear that Australia is no closer to this goal of national uniformity in animal welfare subordinate legislation. However, there is currently no cross-jurisdictional comparison of delegated legislation to provide evidence for this assertion.

Hence, this paper aims to assemble all subordinate laws given authority under the eight state and territory-based animal welfare statutes to understand the extent of the cross-jurisdictional differences. This study follows on the previous statutory comparison [5], by collating the subordinate legislation to provide a more comprehensive understanding of animal welfare laws in Australia, which will help guide future discussions around need for uniformity. Being grounded in law, the detail of this paper is focused only on Australia; however, as a common law country, any discussion points raised throughout the paper are likely relevant to other common law systems that utilize subordinate laws. This paper is designed to provide a state-by-state comparison of the scope of subordinate legislation, adoption of national documents and methods of incorporation into law. It is designed as a reference source for practitioners working in specific areas or with named species to be able to easily identify relevant documents pertaining to welfare and understand the nature of their responsibilities in terms of compliance with these documents. It is not intended to provide a full review on the details of each regulation and code included for analysis. Thus, this review will discuss the different approaches in Australian jurisdictions, disparities between level of protection across states, and potential avenues for reform.

## 2. Materials and Methods

### 2.1. Data Sources

Subordinate legislation in the form of regulations and codes of practice was identified directly from the eight Australian state and territory-based animal welfare statutes (enabling acts) in force between 28 March 2022 and 5 April 2022 (Table 1). Federal legislation was not included as animal welfare is a residual power in Australia, meaning it is in the domain of the states and territories to enact individual legislation [1]. Subordinate legislation accepted for analysis included regulations that were delegated to a Minister by an animal welfare statute (making them enabling acts). Any regulations merely referenced in statute were not included in the analysis. Additionally, codes of practices that were prescribed in an enabling act or regulation or detailed through the relevant government website were included for analysis. Codes are defined as either being compulsory, meaning they are legally enforceable (i.e., noncompliance is an offence), or voluntary, meaning they act as guidance documents (i.e., noncompliance is not an offence).

### 2.2. Eligibility Criteria

All identified subordinate legislation adopted or prescribed (incorporated under a provision in the enabling statute) under the eight state and territory-based animal welfare enabling acts (Table 1) was accepted for analysis. Subordinate legislation had to either be referred to in statute (or corresponding regulation in the case of some codes) or published on a government website (URL ending in ‘gov.au’) for reliability. This paper does not intend to provide a comprehensive review of all subordinate laws pertaining to animals and their welfare, only those the state and territory-based governments have deemed to be appropriate to prescribe under the eight animal welfare statutes. For this reason, some subordinate laws relating to livestock management, pest control, fisheries, domestic animal management, veterinary practice, animals used in research, wildlife conservation and exhibited animal management are not included in this paper as they fall out of the scope of animal welfare statutes and are prescribed under different legislative frameworks.

### 2.3. Data Extraction

Due to the breadth of subordinate legislation, this paper does not review the entirety of each regulation and code; rather, it focuses on the scope or area of protection it awards to animals. Regulations were manually reviewed to extract the type of animal use it pertains to (e.g., breeding animals, cattle management, animals in rodeos) or the process it controls (e.g., surgical procedures, use of electrical devices or traps, culling procedures). Codes were manually reviewed similarly. However, in some cases, this information was not publicly available. In this case, the title of the code and the accompanying explanatory statement with the relevant Minister’s signature at the time were used to confirm the codes scope of protection. Each Australian jurisdiction operates differently; therefore, when areas of animal use are not included in this paper, it should be assumed that their protection does not fall under animal welfare statute rather than that they do not receive any protection whatsoever.

## 3. Results

Using the search strategies stemming from the eight enabling acts, this study identified 18 regulations (Table 2), 79 compulsory codes (see Appendix A) and 104 voluntary codes (see Appendix B) in force. Each piece of subordinate legislation was included for review and allocated into the following utility categories: companion animals, production animals, wild/exotic animals, or animals used in entertainment. The areas of welfare protection discussed in the results are not intended to be an exhaustive list of all areas relating to each area of animal use, as it is recognized that some areas would be regulated under different statutes. However, these are the areas that each Australian jurisdiction has chosen to regulate under the animal welfare legislative framework.

### 3.1. Companion Animal Protection

The use of subordinate legislation for companion animal protection differs between each jurisdiction (Table 3). Whilst ACT is the most comprehensive based on having the most welfare inclusions, NSW arguably has the greatest enforceability through the adoption of compulsory codes, unlike the voluntary codes used in ACT. Additionally, TAS has the inclusion of specialty regulations specific to dog welfare, and therefore, the majority of their companion animal protections (at least pertaining to dogs) are given more weight through their incorporation into regulations rather than codes. On this note, the use of electrical devices and administration of surgical procedures are consistently regulated through statutes or regulations in each jurisdiction; thus, we can assume by their positioning in the regulatory framework that these are the issues to which the states and territories assign the greatest importance. These are closely followed by breeding standards regulation. Due to the systematic approach of reviewing subordinate laws enabled under animal welfare statutes, it should be noted that aspects of companion animal protection are likely contained in the state and territory animal management statutes, which are not depicted below.

### 3.2. Production Animal Protection

Subordinate legislation specific to production animals was generally consistent between the jurisdictions (Table 4), as most states have chosen to regulate through codes (often the CSIRO National Model Codes of Practice [18] or newly formed Australian Animal Welfare Standards and Guidelines when available [20]). However, each jurisdiction has discretion when deciding whether to make the national codes compulsory or voluntary, and thus, each jurisdiction differs in this aspect. New South Wales, SA and WA have the most compulsory codes, providing greater weight to the provisions therein, whilst ACT, NT and VIC have a greater number of voluntary codes. Aspects of production animal protection that are given the most weight under subordinate legislation are pig and poultry farming and livestock transportation, as these areas are regularly incorporated into regulations rather than codes, providing direct enforceability. Each state and territory also regulates production animal management under livestock statutes, which could contain animal welfare provisions that are not depicted below.

### 3.3. Wild/Exotic Animal Protection

Animal welfare subordinate legislation used for the protection of wild animals or exotic species kept in captivity differs substantially throughout the jurisdictions (Table 5). The most regulated area across the states and territories relates to the trapping of animals. Further, these are consistently given the most legal weight through their inclusion in regulations, and in statutes in the case of NSW and TAS. As these results only focused on the animal welfare legislative framework in Australia (not the national parks and wildlife legislative framework), it is likely that many of these provisions are also regulated under wildlife-based statutes.

### 3.4. Animals used for Entertainment

Subordinate legislation used to protect animals used for entertainment purposes shows a lack of consistency across the jurisdictions (Table 6). However, rodeos are almost universally regulated across the states and territories through incorporation into regulations or via a compulsory code. Similarly to the wildlife provisions, many states and territories regulate animals used for entertainment within different statutes (other than the animal welfare statutes), meaning that the gaps in the below table should not be taken to indicate that there is no regulation for those provisions; rather, it shows that those provisions are not regulated by animal welfare laws. Many states have exhibited animal statutes that likely house such provisions.

### 3.5. Offences for Noncompliance with Compulsory Codes

All jurisdictions list their offences for failure to comply with a compulsory code in statute except for SA, which uses regulations (Table 7). Each offence has a monetary maximum penalty attached that is applicable to natural persons. Every jurisdiction aside from SA uses penalty units to describe fines, which are standard units used to calculate the dollar amount of a fine. A separate sentencing statute defines the monetary value of a penalty unit to ensure the value of a fine is in line with public policy and inflation. Each dollar estimate was calculated based on the penalty unit defined as on 6 September 2022, and these should be taken as estimated only and are not reflective of the amounts long term. No jurisdictions list imprisonment as a penalty option for code breach. Whilst NSW has an offence for noncompliance with compulsory codes prescribed in schedule 1 of the *Prevention of Cruelty to Animals Regulation 2012* (NSW), this offence does not apply to the compulsory codes listed under reg 33 as those are only admissible in court, meaning they can only be used as evidence in the circumstances where a defendant was charged with an offence under the enabling animal welfare statute. Similarly, WA compulsory codes are only admissible in court and there are no offences included for noncompliance. Finally, the statutory wording of the Victorian *Prevention of Cruelty to Animals Act 1986* (Vic) suggests that compliance with codes (both compulsory and voluntary) can be used as a defence to an animal welfare offence; thus, there are no direct offences for noncompliance with a code.

### 3.6. Jurisdictional Code of Practice Framework

Every jurisdiction has discretion when deciding which codes are compulsory or voluntary, resulting in differences between the frameworks in each state and territory (Figure 2). Each jurisdiction has variations in how (and which) codes of practices were adopted, and there were no clear observable patterns between jurisdictions. Whilst SA and WA have only compulsory codes prescribed under their animal welfare statute, NT differs by only listing voluntary codes, and the remainder use a combination of both compulsory and voluntary. All jurisdictions (aside from NSW) have a greater number of voluntary codes in force than compulsory codes.

Of all codes included for analysis, those related to production animals were most common (58.3%). Within those production animal codes, the majority were adopted as voluntary (59.8%), whilst the remainder were compulsory (40.2%). Companion animal codes (equating to 16% of all codes) were evenly split between compulsory (53.6%) and voluntary (46.4%), whilst wild/exotic animal codes equated to 16.6% of all codes (compulsory 20.7%; voluntary 79.3%), and codes used for animals in entertainment amounted to 9.2% of all codes (compulsory 68.8%; voluntary 31.2%).

### 3.7. Jurisdictional Acceptance of Animal Welfare Standards

The four currently available animal welfare standards as of 6 September 2022 [20] (transport, cattle, sheep and saleyards/depots) have been variably adopted by the states and territories (Figure 3). It is worth noting that although the majority of jurisdictions have incorporated the standards under the animal welfare legislative framework, NT and VIC have incorporated them under their livestock management legislation. Thus, in order to accurately depict the acceptance rates, these were not excluded from the analysis for the purposes of Figure 3. Whilst the transport standards have been adopted by each jurisdiction, the cattle standards have been adopted by four jurisdictions (50%; NSW, QLD, SA, WA). Similarly, the sheep standards are currently adopted in three jurisdictions (37.5%; NSW, QLD, SA) and the saleyard and depot standards have a 25% acceptance rate with only two states adopting them under legislation (QLD, WA). These acceptance rates are in accordance with the date of finalization for each of the standards, with the transport standards released in 2013 [46], cattle and sheep in 2016 [47,48] and saleyards and depot most recently released in 2018 [49].

## 4. Discussion

This paper outlines the scope of animal welfare protection detailed in subordinate legislation enabled under the state-based animal welfare statutes in Australia. Following from our previous cross-jurisdictional comparison of animal welfare statutes [5], this analysis is intended to be complementary and to provide a more complete picture of the animal welfare law framework by including subordinate laws in that paradigm. Although grounded in Australia’s domestic law, the findings from this study likely have relevance to other countries that adopt a common law system, which could be the subject of broader international research. Firstly, the differences in quantity of primary (statute) and secondary (subordinate) legislation are substantial, with the 201 subordinate laws identified from this study equating to 96.2% of all sources of laws within the animal welfare law framework. Although animal welfare statutes are crucial, as they commonly deal with issues of animal cruelty or duty of care breaches [50], subordinate laws provide the detail on the range of human–animal interactions that occur in everyday life, arguably encompassing the more prevalent animal welfare issues in society since they relate to everyday husbandry practices and procedures. Recognizing this importance, the remainder of this paper will discuss selected aspects of our analysis with a focus on the current extent of uniformity, and potential avenues for reform.

### 4.1. The Current State of Uniformity

#### 4.1.1. The Extent of Uniformity

Although this paper has identified some cross-jurisdictional differences between the areas of welfare protection covered by subordinate legislation, there were also striking similarities. For example, the states and territories are fairly consistent in giving the most legal weight (determined based on their regulation of the area using documents sitting higher in the legal hierarchy) to similar issues of welfare concern. This was observed for the use of electrical devices and the performance of surgical procedures in companion animals, pig and poultry farming and livestock transportation. Whilst we cannot know the reasoning behind legislators’ and government officials’ decisions to place these provisions in higher-order legislative instruments, we can reason that it may relate to perception of threat to welfare, thus requiring increased enforceability. It is noteworthy that these are often the areas that have come under public scrutiny both domestically and globally, such as the welfare movement to ban battery cages for poultry farming [51,52,53,54] and farrowing crates in piggeries [55,56,57]. This implies that jurisdictions may be prepared to respond to public pressure and global animal welfare trends, although this would likely only come after some form of regulatory impact assessment considering economic impacts amongst other considerations. The fact that each jurisdiction has uniformly adopted the Land Transport of Livestock Standards [58] implies regulators are cognisant of the difficulties in complying with different standards when travelling between jurisdictions [59], which suggest the states and territories will take a pragmatic approach towards uniformity when it makes sense to do so.

Australia is a vast and diverse country, with environmental, economic and social conditions varying across the jurisdictions [60]. Animal welfare legislative frameworks will be a product of the jurisdiction’s locality and associated geography of their land. As one example, livestock production in Australia differs vastly across the states and territories, with NSW, QLD, VIC and WA being the highest producers of red meat [61], poultry [62] and pork [63], whereas ACT and NT are rarely reported in those statistics. Thus, it is not surprising that ACT and NT had the least comprehensive scope of subordinate legislation protecting these industries. The states and territories will only adopt subordinate laws that are relevant for their locality, as this is grounded in the theory of federalism in Australia and the subsequent ‘principle of subsidiarity’ [64]. Subsidiarity is the idea that a decision should be made at the most local level of government where there is shared community interest [65]. Thus, giving the states and territories of Australia discretion to make policy decisions that reflect their own societal pressures and circumstances, allowing for policy to meet local preferences and needs [64]. The theory of federalism specifically warns against unnecessary uniformity between the states [64], as a ‘one size fits all’ approach would ignore the extent of diversity within Australia, reduce response rates to local issues and cause competition with national concerns [60]. Thus, it is likely that a uniform approach might not be necessary for some areas of animal welfare protection when considering the states’ locality.

Conversely, given this observation in low livestock rearing states, it is somewhat surprising that QLD and VIC have the highest proportion of voluntary codes considering they are some of the highest livestock producers. This leaves the details of husbandry and management in these industries outside the bounds of subordinate legislation and in the realms of judicial interpretation via interpretations of ‘cruelty’ as expounded in animal welfare statutes. Thus, although the locality can cause justified inconsistencies based on the state’s locality, there still appears to be a dis-uniform approach to some areas that locality cannot explain, and perhaps political decisions are at play. However, it should be noted that both QLD and VIC are in the processes of reforming their animal welfare legislation, with both jurisdictions proposing to improve the welfare standards of production animals (QLD [66]; VIC [67]). As the bills still require parliamentary debates, at this stage the outcome remains unknown.

#### 4.1.2. Uniform Licensing of Animal Research

An area not depicted in the results of this study was the approach the jurisdictions have taken to regulating the welfare of animals used for scientific purposes. Each jurisdiction has uniformly adopted the latest edition of the Australian Code for the Care and Use of Animals for Scientific Purposes (‘Research Code’) [68]. The incorporation of this code into law generally differs from other Codes (as depicted in Table 7).

Code compliance is generally a license condition rather than a direct provision laid out within the Act. State-based statutes authorize Ministers to issue licenses to institutions for the use of animals in research. In general, these licenses are expected to refer to compliance with the code as being a license condition but noting that some states have maintained some discretionary power in this regard through the act wording, for example in the *Animal Welfare Act 1985* (SA) s 19(2)(f) ‘The Minister *may* impose conditions requiring the holder of the license to comply with *such provisions* of the Code as *may be* specified in the conditions’. Whilst it is suspected that in reality most issued licenses refer to the Code in its entirety, it is clear that there is at least the *potential* for dis-uniformity in Code usage across the states and territories. Research institutions monitor compliance with the code internally through the nomination of an animal ethics committee that approves animal research in accordance with the Research Code. Any research carried out in breach of the Research Code will result in the potential loss of license by the institution and attract higher penalties, sometimes upwards of $10,000 [69].

There is also the niggling question of whether all aspects of the Code are actually enforceable in some states due to lack of coverage in the overarching enabling statute. For example, the Code covers all living vertebrates except human beings and includes fish. However, if fish are not considered within the definition of animals in the primary act enabling the Code, do they fall outside of the scope of protection? Alternatively, by referring to the Code in license provisions, perhaps they can legitimately be protected in research since any Code breach is then technically a license breach, signaling an act breach. This question likely needs further legal examination and judicial interpretation but again serves to highlight issues of dis-uniformity across the states.

Additionally, whilst there may be almost uniform adoption of the Code across the jurisdictions, the mode of adoption does differ. For example, seven jurisdictions (ACT, QLD, SA, TAS, VIC, WA, NT) incorporate compliance with the Research Code through licensing under their animal welfare statute, however NSW differs as they have incorporated it under the *Animal Research Act 1985* (NSW). This raises questions around the interpretation of ‘uniformity’ within this context, as although the animal research code is uniformly functioning with its intended purpose in each jurisdiction (at least based on what the legislation implies), the framework applied is still dis-uniform. This is a common occurrence within animal welfare legislation, as often the scope of animal welfare protection is balanced against other legislative frameworks involving animals.

#### 4.1.3. Uniform Function or Uniform Framework?

The nature of human–animal interactions is diverse and often based on individual and species-based considerations. They vary from our relationships with companion animals, which are often intrinsic in nature [70,71,72], to the utilitarian relationships underpinning the livestock industry [73] as well as management strategies for wildlife conservation [3] and pest control [15], amongst others. Each type of interaction is grounded within different legislative frameworks [5], resulting in a substantial mass of overlapping animal-related legislation. This creates inconsistencies in the animal welfare legislation between the jurisdictions, as is apparent in these results. The question of whether uniformity is required in just function, or within both function and framework is difficult. Arguably the states and territories have already taken a ‘broadly’ uniform approach to legislating the issue of animal welfare functionally [5]; ‘broadly’ in that the majority of the inconsistencies are likely due to local pressures and circumstances. Therefore, although the legal frameworks of animal welfare protection may differ, the mandated provisions are likely similar. However, this cannot be proven without a national understanding of enforcement statistics, but as academic commentary suggests, national data collection is almost impossible without a uniform framework [3]. Hence, an impasse has been reached.

The interpretation that a uniform framework will result in uniform function is too simplistic, as previous research has established that even with uniform statute application dis-uniformity may remain though differences in enforcement [74]. That is, even if a uniform approach were adopted in framework, there is no guarantee that it would be applied similarly and consistently across jurisdictions given the differences between enforcement agencies across the states [8]. The lack of a national data collation system is still a substantial shortcoming within Australian animal welfare law and likely requires some form of national oversight to manage and monitor. This can likely be achieved without any form of uniform legislative framework. However, it would require animal management (which would include animal welfare in that paradigm) to be incorporated under a current federal office (e.g., Department of Agriculture, Fisheries and Forestry/Department of Climate Change, Energy, the Environment and Water) or as Mundt [75] suggested, the development of a new office for animal management. Although this would be a burden on the federal executive arm of government to develop, it could be less burdensome than reshaping the current jurisdictional frameworks to achieve uniformity. With national data collection, the function of the states can be monitored to ensure that Australia as a nation is upholding uniform standards of animal welfare whilst accounting for any cross-jurisdictional differences due to local societal pressures and circumstances. In addition, it would also provide greater transparency within the function of the executive arm of government and their development of subordinate laws, something that is currently very secretive in nature [12].

### 4.2. Outdated, Underused and Conflicting Systems

#### 4.2.1. Subordinate Legislation Is Amended ‘Quickly’ and ‘Easily’

One of the key features of subordinate legislation is how easily and quickly it can be amended to keep in line with public concerns and scientific advancements [12,13]. To be formed, subordinate laws must comply with a process set out in state-based subordinate legislation statutes. However, this process does not require any of the accountability and transparency measures required by the parliamentary law-making process, resulting in little to no available information documenting the consultation process or explaining why the change in subordinate law was necessary [12]. This means although the process is far quicker than statutory changes, it is ultimately much more secret and hidden behind closed doors. There is no way of understanding the types of information the executive government relied on or what influenced their decisions, something that is rather concerning in the context of animal welfare, where decisions often need to be balanced against the profitability of animal industries [76]. Although parliament are given the opportunity to disallow subordinate legislation, the sheer volume of it entering the system detracts from the efficacy of this form of oversight [12,13], meaning that the executive are essentially making laws that govern every day human-animal interactions with minimal, if any, parliamentary oversight.

Furthermore, subordinate laws often date further back than statutes. For example, a majority of the animal welfare codes still in force are underpinned by the Model Codes of Practice that were introduced in the 1980s, whilst the last two decades have seen somewhat frequent amendments to animal welfare statutes throughout the jurisdictions [77,78,79,80,81,82]. Hence, although in theory, amendments to subordinate laws may be quicker and easier, in practice, they appear to be rarely instigated. This raises concerns about their use considering that the dated nature of the codes means they are likely not representing current best animal welfare practice [2].

#### 4.2.2. Conflicts of Interest

A point briefly touched on above is the need for balance between the protection of animal welfare and the profitability of animal-related industries. This is an area that subordinate laws neatly fall within as the Ministers who are delegated authority to create subordinate laws are often those involved in government agricultural departments. This creates a conflict of interest between promoting the profitability of the livestock industry and regulating it for animal welfare protection [2,76]. The main point of discourse is that protecting animal welfare is not always the most profitable option, and it has been suggested that the profitable option holds more sway over the executive [76], potentially resulting in subordinate laws poorly reflecting animal welfare. Again, as the entire process of creating subordinate laws is undisclosed [12], it is not possible to confirm such concern. Greater transparency from the executive is required to dispel this concern.

On a similar note, the consultation processes involved in the development of codes of practice are also alleged to conflict with animal welfare protection. Ideally, the consultation process brings together all stakeholders involved within the relevant animal industry (often livestock industries) to understand each viewpoint and achieve balance between ethical views and practical working arrangements [59]. However, literature has suggested that industry representatives have a disproportionate influence on the decision-making process, as their input often overwhelms the limited input from organizations committed to improving animal welfare [2,13]. Therefore, there is a point at which codes tend to cease protecting animal welfare when it is in conflict with industries’ interests [17]. This suggests that codes are based on an animal’s extrinsic value (worth to humans) rather than their sentient abilities [72]. This is especially concerning, as some jurisdictions (such as VIC) will consider compliance with a code as a defence for an animal cruelty offence [83]. However, it should be noted that some literature suggests otherwise: Edge and Barnett [84] found that during the consultation process of the Animal Welfare Transport Standards, all stakeholders (including representatives from science, welfare, industry and government) had similar beliefs regarding animal welfare and were able to come to a consensus. Hence, again this is likely another issue of transparency within subordinate law development.

### 4.3. Moving Forward with the Current System

#### Is Uniformity Feasible?

Whilst a national uniform approach to animal welfare might be beneficial, it is likely not feasible within the current legal framework given the constitutional restrictions [5]. This is in recognition of the fact that achieving national uniformity to animal welfare legislation is likely very burdensome, and slow progressing, given learnings from other areas of law such as the experience with the Uniform Evidence Acts [74]. Further, it already has a history of failure in Australia given the resource challenges experienced with the previous Australian Animal Welfare Strategy (AAWS) led by the Federal Government [3,17]. This is not to say that uniformity should not be considered or that success is unlikely in the future. However there are some prominent issues identified from this paper that can be addressed within the current framework that do not require national uniformity of animal welfare legislation.

In lieu of a uniform approach, some form of federal database should be implemented in the interim. As previously discussed, national data collection will confirm Morton et al.’s [5] hypothesis that animal protection laws are broadly functioning uniformly across the jurisdictions with cross-jurisdictional differences likely accounted for by locality, rather than opposing standards of animal welfare. In addition, a federal database will provide transparency and accountability within the development of subordinate legislation, as it would provide an initiative to improve subordinate laws in line with other jurisdictions. Indirectly, this would likely reduce the number of out-of-date codes of practice and create a greater level of uniformity in animal welfare subordinate laws simply through allowing the states to regulate themselves rather than completely reforming the whole framework.

Additionally, such a database should be readily available and accessible to animal-related industries, as this will allow industries to understand their expectations in relation to animal welfare, as well as use the subordinate legislation (as many codes of practice were not readily available online during the data collection stage of this research). The database should contain information similar to this paper but hold further in-depth details on each regulation and code of practice written in lay language with the accompanying formal subordinate law attached. Ideally this would have search functionality to easily source and access these data. This tool could assist in improving the value of subordinate laws for informing and educating persons on their obligations to animals rather than just being used to punish noncompliance or providing defences to cruelty offences.

## 5. Conclusions

Animal welfare subordinate legislation in Australia, in the forms of regulations and codes of practices, is much more extensive than some may think, making up the vast majority of all sources of laws within the animal welfare legislative framework. These documents house details on the range of human–animal interactions that occur in everyday life. Given that these laws contain provisions relating to everyday husbandry practices and procedures, it is concerning that a majority were often decades old, such as the Model Codes of Practice, and hence, likely not reflecting the current societally accepted standards for animal welfare, and benchmarks for animal welfare science. However, in terms of uniformity, it was identified that each jurisdiction took a fairly consistent approach by giving the most legal weight to similar topics. This was achieved through incorporation in the regulations. Additionally, it is purported that dis-uniformity is likely caused by local societal pressures and circumstances, a concept that is encouraged through the concept of federalism, as well as overlapping animal-related legislative frameworks. Some form of federal data collation is required to confirm such hypotheses and provide some form of accountability in subordinate law-making in animal law in Australia. Such a database is recommended to be available online and written in lay language to enable animal-related industries to better understand their expectations in relation to animal welfare. Coupled with good evidence-based drafting and consultation, subordinate laws may even provide insight into goals for animal caretakers to strive towards for enhanced welfare protection, to shift from a reactive to a proactive approach to animal welfare.

## Figures and Tables

**Figure 1 animals-12-02437-f001:**
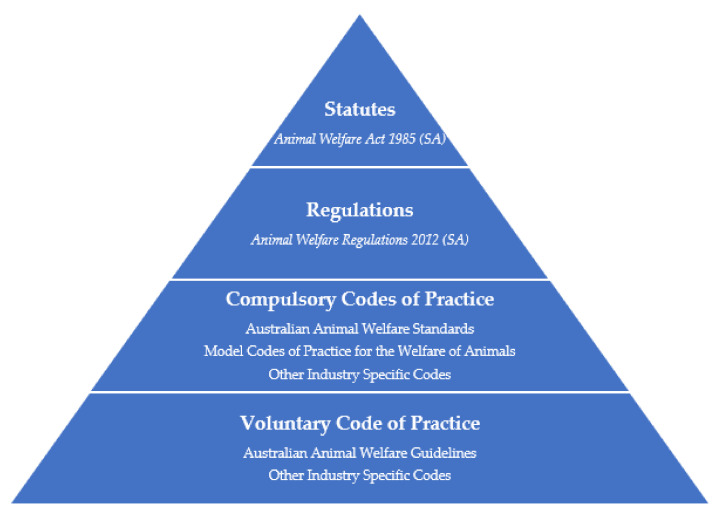
Hierarchy of state and territory legislation in relation to animal welfare protection. Note that the legislation listed use South Australia (SA) as an example and do not represent a comprehensive list.

**Figure 2 animals-12-02437-f002:**
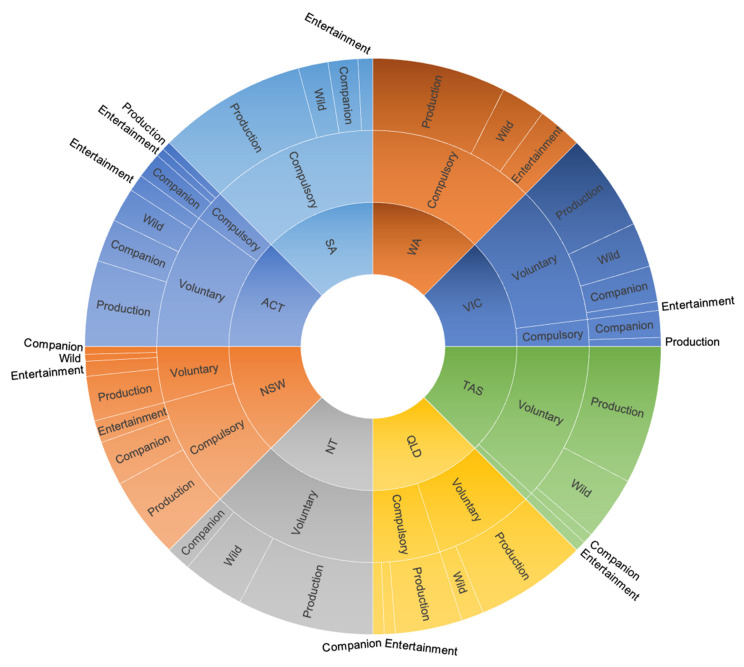
Sunburst diagram of the relative number of compulsory and voluntary codes per each Australian jurisdiction, per utility group. Each color represents a different jurisdiction, with the number of compulsory and voluntary codes prescribed under animal welfare statutes in that jurisdiction. This is broken down further to show use of compulsory or voluntary codes across each animal utility group (companion, production, wild and entertainment).

**Figure 3 animals-12-02437-f003:**
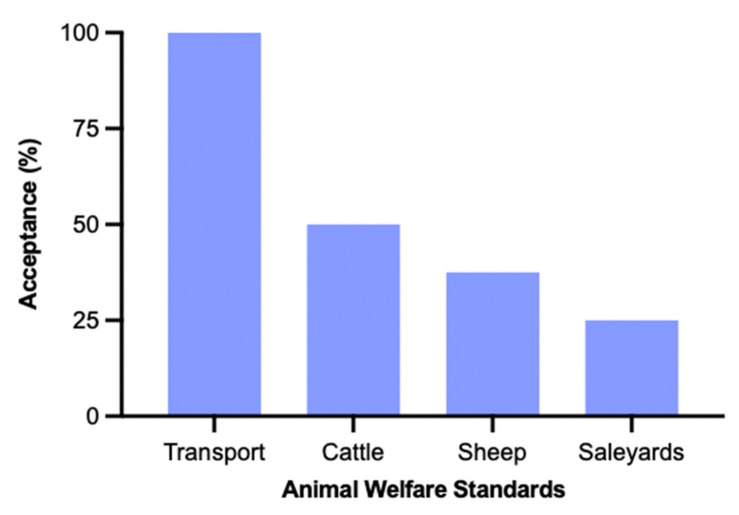
Acceptance rates of the animal welfare standards [20] currently available across the Australian jurisdictions. Acceptance is measured by their incorporation under any legislation (not exclusive to animal welfare legislation), in primary or secondary form. Transport standards have been accepted by all jurisdictions (100%); cattle standards accepted by NSW, QLD, SA, WA (50%); sheep standards accepted by NSW, QLD, SA (37.5%); saleyard and depot standards accepted by QLD, WA (25%).

**Table 1 animals-12-02437-t001:** Enabling acts for each animal welfare subordinate legislation included for analysis.

Jurisdiction	Enabling Act
Australian Capital Territory (ACT)	*Animal Welfare Act 1992* [21]
New South Wales (NSW)	*Prevention of Cruelty to Animals Act 1979* [22]
Northern Territory (NT)	*Animal Welfare Act 1999* [23]
Queensland (QLD)	*Animal Care and Protection Act 2001* [24]
South Australia (SA)	*Animal Welfare Act 1985* [25]
Tasmania (TAS)	*Animal Welfare Act 1993* [26]
Victoria (VIC)	*Prevention of Cruelty to Animals Act 1986* [27]
Western Australia (WA)	*Animal Welfare Act 2002* [28]

**Table 2 animals-12-02437-t002:** Subordinate legislation in the form of regulations made under animal welfare statutes for each Australian state and territory.

Jurisdiction	Regulation
ACT	*Animal Welfare Regulation 2001* [29]
NSW	*Prevention of Cruelty to Animals Regulation 2012* [30]
NT	*Animal Welfare Regulations 2000* [31]
QLD	*Animal Care and Protection Regulation 2012* [32]
SA	*Animal Welfare Regulations 2012* [33]
TAS	*Animal Welfare (Dogs) Regulations 2016* [34] *Animal Welfare (Domestic Poultry) Regulations 2013* [35] *Animal Welfare (General) Regulations 2013* [36] *Animal Welfare (Land Transport of Livestock) Regulations 2013* [37] *Animal Welfare (Pigs) Regulations 2013* [38]
VIC	*Prevention of Cruelty to Animals Regulations 2019* [39] *Prevention of Cruelty to Animals (Domestic Fowl) Regulations 2016* [40]
WA	*Animal Welfare (Commercial Poultry) Regulations 2008* [41] *Animal Welfare (General) Regulations 2003* [42] *Animal Welfare (Pig Industry) Regulations 2010* [43] *Animal Welfare (Scientific Purposes) Regulations 2003* [44] *Animal Welfare (Transport, Saleyards and Depots) (Cattle and Sheep) Regulations 2020* [45]

**Table 3 animals-12-02437-t003:** Areas of welfare protection assigned to companion animals via subordinate legislation in each Australian jurisdiction. The terms ‘compulsory’ and ‘voluntary’ refer to the status of the code of practice. For a detailed list of all compulsory and voluntary codes, refer to, Appendix A and Appendix B respectively. This table is not intended to be an exhaustive list of all areas of companion animal protection, just of those areas that were identified from the search strategies.

Welfare Protection	Jurisdiction							
	ACT	NSW	NT	QLD	SA	TAS	VIC	WA
Boarding	Compulsory	Compulsory				Regulations (dog)		
Breeding	***	Compulsory		Compulsory	Compulsory	Regulations (dog)	Compulsory	
Dog daycare	Compulsory					Regulations (dog) *		
Electrical devices	Regulationsreg 5A	Regulations reg 35	***		Regulationsreg 8(1)(a)	Regulationsreg 8 ***	Regulations reg 14(1) & Compulsory	Regulations (general) reg 3(a)
Grooming	Voluntary	Compulsory						
Keeping birds	Voluntary	Compulsory	Voluntary		Compulsory		Voluntary	
Keeping cats	Voluntary						Voluntary	
Keeping dogs	Voluntary					Regulations (dog)	Voluntary	Compulsory
Pet shops	Compulsory	Compulsory	Voluntary		Compulsory			
Sale of animals	Compulsory***				Compulsory			
Shelters/pounds	Voluntary				Compulsory	Regulations (dog)		
Surgical procedures	***	Regulations Part 3		Regulations Part 4 ***	Regulations reg 6(1)(a)		***	***
Transport	Regulation reg 15A	Compulsory				Regulations (dog) reg 14	Regulations reg 6	
Working dogs		Compulsory & Voluntary **						

* Does not explicitly state daycare; however, likely daycare would fall under the definition of ‘facility’. ** Security dogs are compulsory; assistance and farm working dogs are voluntary. *** Included in animal welfare statute.

**Table 4 animals-12-02437-t004:** Areas of welfare protection assigned to animals used for farming purposes via subordinate legislation in each Australian jurisdiction. The terms ‘compulsory’ and ‘voluntary’ refer to the status of the code of practice. For a detailed list of all compulsory and voluntary codes, refer to, Appendix A and Appendix B respectively. This table is not intended to be an exhaustive list of all areas of production animal protection, just of those areas that were identified from the search strategies.

Welfare Protection	Jurisdiction							
	ACT	NSW	NT	QLD	SA	TAS	VIC	WA
Buffalo		Compulsory	Voluntary	Voluntary	Compulsory			Compulsory
Captive bred emus		Voluntary	Voluntary	Voluntary	Compulsory	Voluntary	Voluntary	Compulsory
Cattle	Voluntary	Compulsory		Compulsory	Regulations Part 8	Voluntary	Voluntary	Compulsory
Cattle feedlots		Compulsory	Voluntary	Voluntary	Regulations reg 72		Voluntary	
Deer	Voluntary	Compulsory	Voluntary	Voluntary	Compulsory	Voluntary	Voluntary	Compulsory
Goats	Voluntary	Compulsory	Voluntary	Voluntary	Compulsory	Voluntary	Voluntary	Compulsory
Horses	Voluntary					Voluntary	Voluntary	
Ostriches		Voluntary	Voluntary	Voluntary	Compulsory			
Pigs	****	Compulsory & Voluntary	Voluntary	Compulsory	Regulations Part 6	Regulations (pigs) & Voluntary		Regulations (pigs) & Compulsory
Poultry	Regulations Part 6 * & Voluntary	RegulationsPart 2 & Compulsory	Voluntary	Compulsory	Regulations Part 5	Regulations (poultry) & Voluntary	Regulations (fowl) & Voluntary	Regulations (poultry) & Compulsory
Rabbits		Voluntary	Voluntary	Voluntary	Compulsory	Voluntary	Voluntary	Compulsory
Saleyards/depots	Voluntary	Compulsory	Voluntary	Compulsory	Compulsory	Voluntary	Voluntary	Regulations (saleyards) & Compulsory
Sheep	Voluntary	Compulsory		Compulsory	Regulations Part 9	Voluntary	Regulations r 8 & Voluntary	Compulsory
Slaughtering establishments	Voluntary	Voluntary	Voluntary	Voluntary	Compulsory			Compulsory
Transport	Compulsory	Compulsory		Compulsory	RegulationsPart 7 **	Regulations (transport) ***	Compulsory	Regulations (transport) & Compulsory

* Only for egg production; ** SA has additional codes for air and sea transport; *** VIC has a voluntary code for sea transport; **** Included in animal welfare statute.

**Table 5 animals-12-02437-t005:** Areas of welfare protection assigned to wild and exotic animals in captivity via subordinate legislation in each Australian jurisdiction. The terms ‘compulsory’ and ‘voluntary’ refer to the status of the code of practice. For a detailed list of all compulsory and voluntary codes, refer to, Appendix A and Appendix B respectively. This table is not intended to be an exhaustive list of all areas of wild animal protection, just of those areas that were identified from the search strategies.

Welfare Protection	Jurisdiction							
	ACT	NSW	NT	QLD	SA	TAS	VIC	WA
Amphibians/Reptiles	Voluntary						Voluntary	
Camels		Voluntary	Voluntary	Voluntary	Compulsory			Compulsory
Crocodiles			Voluntary					
Feral livestock animals (culling)			Voluntary	Voluntary	Compulsory			Compulsory
Humane pest control	Voluntary		Voluntary			Voluntary		
Hunting/game parks		*				Voluntary	Voluntary	
Recreational fishing	Voluntary							
Rehabilitation							Voluntary	
Traps	Regulations reg 5A & 7C	*			Regulations reg 9(1)	*	Regulations Part 3	Regulations (general) reg 3B

* Included in animal welfare statute.

**Table 6 animals-12-02437-t006:** Areas of welfare protection assigned to animals used for human entertainment via subordinate legislation in each Australian jurisdiction. The terms ‘compulsory’ and ‘voluntary’ refer to the status of the code of practice. For a detailed list of all compulsory and voluntary codes, refer to, Appendix A and Appendix B respectively. This table is not intended to be an exhaustive list of all areas of entertainment animal protection, just of those areas that were identified from the search strategies.

Welfare Protection	Jurisdiction							
	ACT	NSW	NT	QLD	SA	TAS	VIC	WA
Circuses	Regulations Part 4				Compulsory		Voluntary	Compulsory
Exhibited animals							Voluntary	Compulsory
Films/Theatrical performances	Voluntary	Compulsory					Voluntary	
Greyhounds	Compulsory & Voluntary							
Horse riding/racing	Voluntary	Compulsory					Voluntary	Compulsory
Rodeos		Compulsory		Compulsory	Regulations Part 4	Compulsory	Regulations Part 4	Compulsory

**Table 7 animals-12-02437-t007:** Each statutory offence for noncompliance with compulsory codes of practice and the corresponding maximum penalties per each jurisdiction’s animal welfare statute. Penalties are applicable to natural persons. Penalty units refer to a standard unit used to calculate the dollar amount of a fine in line with inflation. Each dollar estimate is based on the penalty units detailed in the crime/sentencing statutes in each jurisdiction as of 6 September 2022. They should be taken as estimates only and will be subject to frequent changes.

Jurisdiction	Offence for Noncompliance	Maximum Penalty
ACT	Animal Welfare Act 1992 Sections 24A-D	100 penalty units (AUD 22,200 estimate)
NSW	Prevention of Cruelty to Animals Act 1979 Section 26(3)(i) *	50 penalty units (AUD 5500 estimate)
NT	None included	
QLD	Animal Care and Protection Act 2001 Section 15(3)	300 penalty units (AUD 43,125 estimate)
SA	Animal Welfare Regulations 2012 Section 5	AUD 2500
TAS	Animal Welfare Act 1993 Section 11A	50 penalty units (AUD 9050 estimate)
VIC	None included **	
WA	None included ***	

* Compulsory codes prescribed under s 34A(1) are only admissible in court; ** Vic does not have offences for noncompliance but rather has basic cruelty offences listed in statute with a statement that by following the compulsory code, a person will not breach a cruelty offence; *** WA compulsory codes are only admissible in court.

## Data Availability

All data are contained within this paper.

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
