# Peer review of "Understanding Subordinate Animal Welfare Legislation in Australia: Assembling the Regulations and Codes of Practice"

_animals, 2022, doi:10.3390/ani12182437_

Round 1

Reviewer 1 Report

The paper presents an assessment of animal welfare legislation in Australia. Much of the study focuses on a critique of the federalist form of government wherein different states have different welfare related legislation and policies resulting in considerable "dis-uniformity. A strong distinction is made between "subordinate legislation" and more binding legislation. 

My feeling is that this very long paper has a very narrow focus and provides too much detail -- e.g., each of the 8 states' subordinate legislations is described when a generalized description would suffice. I would suggest in lieu of so much in the weeds description that a comparison be presented between Australia's political system and those of other countries such as UK, US, and Canada. Also, the literature review might reference recent literature on political theory and science as it relates to our treatment of animals (e.g., Anderson and Kymlicka's Zoopolis.

Author Response

My feeling is that this very long paper has a very narrow focus and provides too much detail -- e.g., each of the 8 states' subordinate legislations is described when a generalized description would suffice.

Thank you for your feedback. We have removed the subsection which detailed all the state subordinate legislation in the methodology section and left a generalised description of our search criteria.

I would suggest in lieu of so much in the weeds description that a comparison be presented between Australia's political system and those of other countries such as UK, US, and Canada.

As each of these countries also adopt a common law system like Australia, although the details of our findings are not relevant, the frameworks that we discussed would be relevant. Each of these countries would likely incorporate subordinate legislation within their animal welfare legislative framework, thus it is likely any issue we identified could be found internationally as well. We have included discussion on lines 118-119, 145-147, 461-463 to make this international relevance clearer.

Also, the literature review might reference recent literature on political theory and science as it relates to our treatment of animals (e.g., Anderson and Kymlicka's Zoopolis.

Thank you for providing us this literature. We have opted not to include it in the discussion of this paper given is it heavily focused on animal rights and changing the political agenda for this rights approach. Our concern is that in order to discuss this political theory appropriately, we would need to address the current legal status of animals (ie. property), which would deviate from our research question of focusing on subordinate laws. We believe such a discussion would be best suited in a review paper where the controversies and appropriateness of how we regulate animal welfare can be discussed in greater detail. Additionally, given our other reviewer report claims the discussion has too many discussion points, we believe that adding another discussion on political theories would create greater disparity within our discussion. Thus, in order to balance our peer review reports, we thank you for your suggestion but have not included it in the paper.

Reviewer 2 Report

A number of questions for the authors to consider when revising:

1.      The section of 3.7. Jurisdictional Acceptance of Animal Welfare Standards is not very clear. It does not indicate whether the info is up to date or not or it is from second sources?

2.       Table 7 should indicate the actual penalties in dollar terms, not just penalty units as most people would not know what they mean.

3.       Re 3.4. Animals used for Entertainment, the section is problematic and it affects the  whole paper. In some states, there are laws for animals used for entertainment instead of codes, so  table 6 and the section are somewhat misleading as if some of the jurisdictions had no protection for such animals.

4.       Figure 2, more explanation or better chart is needed. 

5.       Re method and data extraction, how would it be different from doing it manually, searching and identifying the relevant regulations and codes and standards through each jurisdiction in the old fashion way?  The paper does not claim to be exhaustive anyway and does not study all the relevant regulations, codes etc. so, the research method does not seem to matter much irrespective how the documents were identified. There is no real scholarly value or innovation in the method which can be stated in very simple terms, instead of making it look more than it is. It is understood that people like to use all sorts of colourful graphs and tables these days with all the bells and whistles. Thus, it is suggested that the paper focus more on the analysis of the regulations and codes instead of the method.

6.       The discussion section has too many sub-sections and they are all very brief. Perhaps it  should focus on a limited area  and provide an  indepth discussion.

Author Response

The section of 3.7. Jurisdictional Acceptance of Animal Welfare Standards is not very clear. It does not indicate whether the info is up to date or not or it is from second sources?

All the information was up to date at the time of writing the paper, which we referenced in the figure legend at ref [20]. However, to make this clearer we have added ‘as of 6 September 2022 [20]’ on line 435 as well.

Table 7 should indicate the actual penalties in dollar terms, not just penalty units as most people would not know what they mean.

Thank you for this suggestion. We have included the dollar estimate the penalty units would equate too as of 6 September 2022, and added a discussion on lines 386-388, and in the table legend stating that this information only reflects this current point in time as the dollar amount of penalty units regularly fluctuate.

Re 3.4. Animals used for Entertainment, the section is problematic and it affects the  whole paper. In some states, there are laws for animals used for entertainment instead of codes, so  table 6 and the section are somewhat misleading as if some of the jurisdictions had no protection for such animals.

Our paper only reviewed subordinate laws prescribed under the animal welfare statutes (laws with the primary object of protecting animal welfare), as it is impossible to capture every element of animal regulation, which we included disclaimers addressing this limitation on lines 269-273, 284-287, 293-297. With regards to statutes for animals used for entertainment, the objective of the statute is not focused solely on animal welfare protection, thus meaning that any animal welfare provisions included would be indirect given the objective of these laws, which does not satisfy our research question. However, to make this point clearer we have also added disclaimers on each of the results subsections for companion (lines 35-319), production (line 339-341), wildlife (lines 356-359) and entertainment (lines 367-374).

Figure 2, more explanation or better chart is needed.  

We have amended the figure legend to provide more explanation of the diagram. See lines 431-433.

Re method and data extraction, how would it be different from doing it manually, searching and identifying the relevant regulations and codes and standards through each jurisdiction in the old fashion way?  The paper does not claim to be exhaustive anyway and does not study all the relevant regulations, codes etc. so, the research method does not seem to matter much irrespective how the documents were identified. There is no real scholarly value or innovation in the method which can be stated in very simple terms, instead of making it look more than it is. It is understood that people like to use all sorts of colourful graphs and tables these days with all the bells and whistles. Thus, it is suggested that the paper focus more on the analysis of the regulations and codes instead of the method.

Thank you for your suggestion. This study applied a systematic-type methodology in order to allow transparency in our search criteria and repeatability of our results. This is why we decided to report our methodology fully, however in order to make the methodology more concise we have removed all the information on the state-based search strategies in section 2.2.

The discussion section has too many sub-sections and they are all very brief. Perhaps it  should focus on a limited area  and provide an  indepth discussion

Thank you for the suggestion. We broke our discussion up into two discussion points, being 4.1 uniformity and 4.2 the practical issues with subordinate laws (where 4.3 we discuss future research/recommendations based on our findings). However, we subheaded these sections to make it easier for readers to find information and follow the direction of the discussion, as each of the subheadings are connected to the overall heading (either 4.1 or 4.2). Additionally, as this paper is a primary study there is a presumption that we all discuss all the results rather than a review paper which may have one direct focus. The other review reports have suggested that the discussion is too detailed. In order to balance the conflicting review reports, we have removed some subheadings in avoid breaking up our discussion, the subheadings which were removed are 4.3.2 (federal database discussion) and 4.1.2 (locality discussion).

Round 2

Reviewer 1 Report

I am OK with accepting the paper as revised.